# The New Role of the Dental Assistant and Nurse in the Age of Advanced Artificial Intelligence in Telehealth Orthodontic Care with Dental Monitoring: Preliminary Report

Jana Surovková *[ID], Sára Haluzová, Martin Strunga [ID], Renáta Urban [ID], Michaela Lifková [ID] and Andrej Thurzo *[ID]

Department of Orthodontics, Regenerative and Forensic Dentistry, Faculty of Medicine, Comenius University in Bratislava, 811 02 Bratislava, Slovakia
* Correspondence: surovkova6@uniba.sk (J.S.); thurzo3@uniba.sk (A.T.)

**Abstract:** This paper explores the impact of Artificial Intelligence (AI) on the role of dental assistants and nurses in orthodontic practices, as there is a gap in understanding the currently evolving impact on orthodontic treatment workflows. The introduction of AI-language models such as ChatGPT 4 is changing patient-office communication and transforming the role of orthodontic nurses. Teledentistry is now heavily reliant on AI implementation in orthodontics. This paper presents the proof of a novel concept: an AI-powered orthodontic workflow that provides new responsibilities for an orthodontic nurse. It also provides a report of an assessment of such a workflow in an orthodontic practice that uses an AI solution called Dental Monitoring over a period of three years. The paper evaluates the benefits and drawbacks of daily automated assessments of orthodontic treatment progress, the impact of AI on personalized care, and the new role of a dental assistant. The paper concludes that AI will improve dental practice through more precise and personalized treatment, bringing new roles and responsibilities for trained medical professionals but raising new ethical and legal issues for dental practices.

**Keywords:** dental nurse; dental monitoring; AI; telemedicine; teledentistry; chatGPT; communication; AI-telehealth; quality assessment; practice; dentistry; personalized care; ethical issues

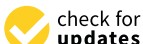



## 1. Introduction

The current role of dental assistants is dynamically changing under the pressure of innovative technologies entering clinical practice. Artificial intelligence (AI)-assisted diagnosis and treatment planning is just one of many trends that are expanding medical scenarios and increasing work efficiency and accuracy [1]. AI provides efficient ways to clinically evaluate radiographic data and support effective treatment planning, representing a significant evolution from analogue processing of cephalometric data [2] towards a 3D digital AI processing era [3–6].

A study by Cheng et al. (2022) provided insights into the path mechanisms underlying healthcare workers' adoption of AI-assisted diagnosis and treatment [7]. Treatment planning is not the domain of dental assistants or nurses, although communication with patients is their most important pillar. Today's world is facing major functional challenges that are leading to the introduction of digital communication tools. Nurses are critical healthcare professionals, and the study by Hellzén et al. 2022 confirms that the development of digital communication in healthcare is critical [7]. With the current empowerment of advanced generative AI such as ChatGPT (OpenAI, San Francisco, CA, USA), built on top of OpenAI's GPT−3.5 family of large language models, person-centered patient care based on digital communication moves to another dimension. The third area of new technologies is the virtual reality environment, with its enormous simulation and therapeutic possibilities. Most care simulation programs focus on the health needs of people in hospitals, and little is known about how to identify these in the home environment [7].

Dental Monitoring® (DM) (Dental Monitoring Co., Paris, France) is used in patients' home environments, and communication between nurse and patient does not theoretically always require nurses' physical presence in the dental office. The study by Yoshioka-Maeda et al. (2022) highlights the importance of developing and validating a virtual reality simulation program for nursing students and the importance of education in this area [8]. Currently, it is possible to easily and effortlessly train a personalized AI using AI technology to create one's own AI module easily and effortlessly (e.g., www.personal.ai accessed on 21 April 2023), which would perfectly mimic the way the nurse or assistant communicates based on the knowledge you would provide, with the possibility of specialized training. Such a virtual meta-human or chatbot, trained on extensive datasets of, for example, nurse-patient communication, could become an effective helper in everyday clinical care, supporting the nurse or dental assistant and preserving their capacity for more complex problems. In dental applications of AI technologies, patients frequently showed a positive attitude towards AI in dentistry [9–11].

Digital technology in the dental field has created new opportunities for orthodontists to connect their practices and collect patient data through teledentistry. Dental Monitoring® (DM) (Dental Monitoring Co., Paris, France) is an orthodontic software that uses AI and knowledge-based algorithms to provide accurate treatment tracking. Developed in Paris by CEO Philippe Salah, it is the first SaaS for dental treatment monitoring. DM's device is programmed to take control of positive movements, and AI software analyzes the aligners' fit and retention to ensure proper biomechanics. It improves interaction between the clinician and the patient and is suitable for all clear orthodontic treatment modalities, even in critical situations, for cases with gold standard compliance and correct device use [10–12].

Telemedicine enhances communication between specialists, for example in cases where a multidisciplinary approach is required, and allows patients to quickly refer to a specialist for a preliminary consultation, clinical follow-up, or follow-up treatment [13]. Telemonitoring aims to replace face-to-face visits with virtual visits for regular monitoring of treatment results and disease progression [14]. Mobile applications and technology play an increasingly important role in daily life and the daily practice of orthodontics. With the number of orthodontic-related applications continuing to increase and the rapid development of artificial intelligence, the potential for enormous benefits for physicians and patients is evident [15]. With the exponential growth of AI and communications technology, the increasing feasibility and remote monitoring of orthodontic treatment has the potential to bring enormous benefits to doctors and patients [11,16].

Figure 1 shows a schematic of the dental nurse's workflow in scenarios with and without the use of Dental Monitoring® (DM) (Dental Monitoring Co., Paris, France) in a specialized orthodontic clinic.

The demand for immediate and invisible treatment alternatives for adults has increased. Therefore, treatment protocols using clear aligners have evolved in response to this demand. Adequate clinical follow-up is critical and can become cumbersome with an increasing frequency of aligner changes. Clinical monitoring with a patient-managed software program is helpful in such cases.

Orthodontists have been encouraged to move forward using tele-orthodontic devices, a term used for remote orthodontic treatment through mechanical means without direct contact with each personal contact [17].

Clinical tracking of aligner-mediated tooth movement is monitored with a patient-managed smartphone application so that the smallest orthodontic movement errors can be detected and corrected early. Such errors would have been difficult to detect given the speed of aligner replacement. Clinical follow-up with a patient-managed smartphone application could thus allow faster and easier control of orthodontic treatment with clear aligners [18].

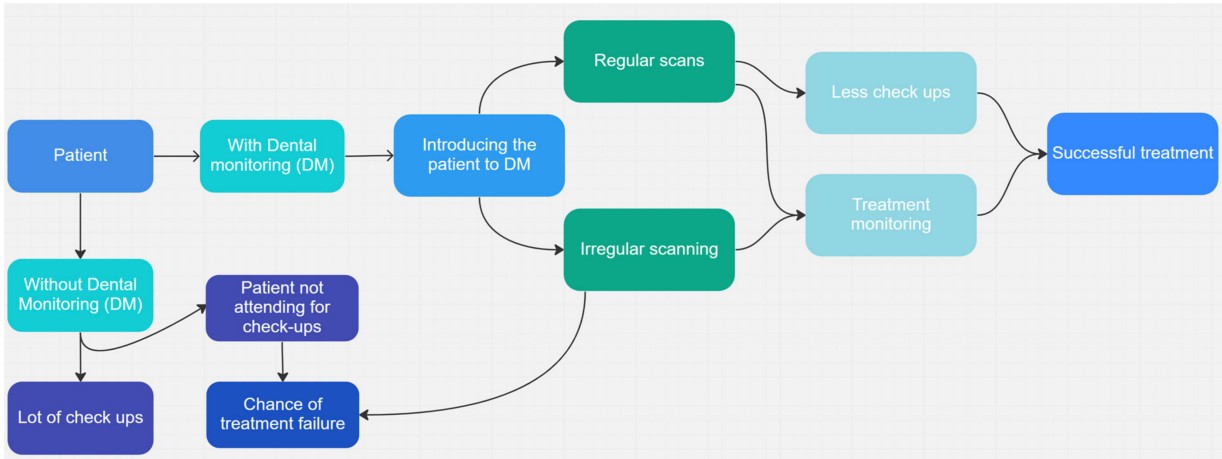

**Figure 1.** Scheme of a dental nurse's workflow with and without the Dental Monitoring® (DM) (Dental Monitoring Co., Paris, France) application.

Dental Monitoring® (DM) (Dental Monitoring Co., Paris, France) has invented new technology to evaluate 3D tooth position from video-scan taken by a patient's smartphone and by using original 3D model from intraoral scan it is able to reconstruct intraoral 3D situation [19].

The effects of Invisalign treatment with transparent aligners with and without Dental Monitoring® (DM) (Dental Monitoring Co., Paris, France) were compared in terms of treatment duration, number of appointments, refinements, refinement aligners, and Invisalign accuracy in achieving predicted tooth positions (aligner tracking). DM with Invisalign therapy resulted in a 3.5 visit (33.1%) reduction in the number of appointments. DM also achieved clinically comparable accuracy in achieving predicted tooth movement compared to the control group in 1.7 fewer months, indicating gradual aligner tracking in the DM group [20].

The last decade has seen a step forward in the evolution of technology with the development of artificial intelligence, which is attracting the attention of researchers worldwide. Every discipline has enthusiastically embraced artificial intelligence, and the field of dentistry is no exception. The increasing demand for patient-related statistics and data requires intelligent software programs to compile and store this information. From the basic step of recording a patient's history to processing the data and extracting information from the data for diagnosis, artificial intelligence has many applications in dentistry and medicine. By no means can artificial intelligence replace the role of the dentist. However, it is important to understand how this technological advance can be used to improve dental practice in the future [21].

The novel coronavirus that causes COVID−19 is transmitted primarily through person-to-person contact but can also be spread through aerosol-generating procedures, so many clinics have very limited person-to-person interactions. The purpose of this paper is to provide useful information to orthodontists, dental assistants, nurses, and office managers considering some form of telemedicine [22,23].

The adaptation of the education curriculum of future nurses needs to respond to this shift in paradigm. Since the era of Web 2.0 systems, online education has changed significantly [24]. Healthcare workers are already under increasing pressure from overwhelming digital communication from patients and healthcare systems, and under pressure, they tend to use limited resources more efficiently [25]. AI can be their empowering tool for more effective and personalized communication as well as more individualized and effective online education.

Digital monitoring has several advantages over analogue impressions. It allows more precise measurements of distances and reduces working time, as the dental technician no longer must request plaster casts [26]. This results in cost reductions for both the laboratory and the patient. The digital workflow can be completely managed by a single person, which means further time savings [27]. In some studies, the use of the DM with the self-ligating technique has reduced the number of appointments for each patient from three appointments in 10 weeks to two appointments [28].

The rationale of this paper is to introduce the concept of a functioning digital workflow, using an AI-powered telehealth solution called Dental Monitoring® (DM) (Dental Monitoring Co., Paris, France) as an important tool in the orthodontic practice. The paper is important as it maps the current and future impact of rapid AI implementation in prevalent digital orthodontic care solutions and necessary adaptations in the roles of dental assistants and nurses.

This study aims to fill the gap in understanding the changing professional responsibilities of dental assistants and nurses, as well as new legal and ethical challenges. The study also contributes to the profession by reporting on current AI telehealth limitations and near future perspectives relevant to the professional development of orthodontic assistants' and nurses' skill sets.

## 2. Materials, Methods, and Concept

### 2.1. Materials

Current basic hardware/technologies and gadgets presented in this paper are shown in Figure 2:

1. The dental monitoring scan box;
2. Retractor;
3. The patient's smartphone.

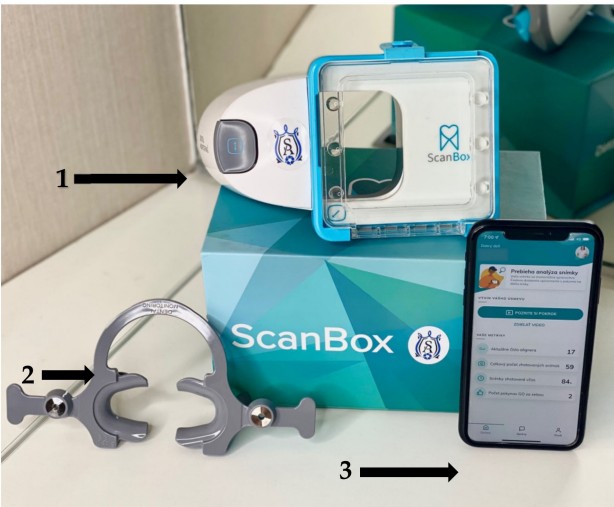

**Figure 2.** The Dental Monitoring® (DM) (Dental Monitoring Co., Paris, France) set up—from the left is the retractor, the middle top is the dental monitoring scan box, and the right is the patient's phone.

The hardware shown in Figure 2 represents the tools necessary for telehealth applications (Dental Monitoring® (DM)) (Dental Monitoring Co., Paris, France) to record an intraoral situation in a patient's home. What is central is that the software, which is powered by AI, can recognize various parameters from video datasets. With the age of AI technology and telecommunications, the healthcare system is changing rapidly. Several telecommunications systems have been applied to hospitals, and over time, a new term, "telemedicine" now represents a real thing in clinical practice [29].

*2.2. Methods*

2.2.1. Method of Evaluation

To evaluate the effect of personalized orthodontic treatment with Dental Monitoring® (DM) (Dental Monitoring Co., Paris, France) in this paper, the following criteria were used:

1.  A comparison of average treatment refinements per patient from a 3 year average before implementation of DM and after. Implementation of DM into a treatment workflow has been compulsory since 2019;
2.  A reduction in the number of in-person appointments required for patients who received dental monitoring and in the period without DM;
3.  An online patient satisfaction feedback provided after every appointment; an improvement in patient satisfaction and comfort level with the use of Dental Monitoring® (DM) (Dental Monitoring Co., Paris, France);
4.  Evaluation of the nurse's/assistant's daily activities in relation to the organization of time.

Only comprehensive aligner treatments longer than a year were selected for evaluation to prevent possible bias resulting from extremely short treatments. This preliminary report refers to preliminary findings on the sets of 372 DM patients and 192 non-DM patients (control), although the complete retrospective cohort study is the subject of separate research and is not pivotal to this preliminary report. The average number of refinements per patient in orthodontic aligner treatment can vary depending on the specific case and treatment plan. In general, it is common for patients to have approximately two refinements during their treatment with aligners. However, this depends on practitioners' and patients' attention to detail and many other factors. Some cases may require more adjustments, while others may not.

Only aligner treatments by Invisalign (Align Technology, Inc., San Jose, CA, USA) were included as they represented the majority of aligner cases treated in the observed period in selected offices. Spark cases (Ormco Corporation, Brea, CA, USA) were excluded.

The time periods evaluated were non-DM workflow 1/2016–1/2019 and DM 1/2019–1/2022 with an approximately stable patient count per year and staff numbers.

An online patient satisfaction survey was provided after every appointment to ascertain the improvement in patient satisfaction and comfort level with the use of Dental Monitoring® (DM) (Dental Monitoring Co., Paris, France). The online form was provided in an electronic appointment reminder and calendar event as a link to an online form created in Typeform.com (Typeform S.L., Barcelona, Spain).

A more complex cost-benefit analysis of Dental Monitoring® (DM) (Dental Monitoring Co., Paris, France) compared to traditional orthodontic treatment methods would require separate research and is a possible topic for a separate clinical trial that would demonstrate the efficacy and safety of Dental Monitoring® (DM) (Dental Monitoring Co., Paris, France) as a means of delivering personalized orthodontic care.

2.2.2. Practical Aspects of Dental Monitoring® (DM) (Dental Monitoring Co., Paris, France)

Patients who do not use Dental Monitoring have to go for regular check-ups, which means that they always get a certain number of aligners, which they wear until the time of the check-up. Usually, this is a month, maximum two. During this time, we have minimal interaction with the patient; we do not see if there is a problem or if the treatment is going as planned. If a problem does arise, the doctor usually does not discover it promptly enough, so the treatment must be adjusted or redone. The patient also had to check for the presence of attachments and report if they were dropping off. Many times, the patient himself does not notice the loss of the attachment.

In the monitored orthodontic practice, 98% of patients are users of the DM app. The advantage of DM is the reduced frequency of visits to the outpatient clinic. The use of DM eliminates frequent check-ups, where many times no problem is found or, on the contrary, the doctor states that treatment has failed. Patients without DM were given aligners for a while until the next check-up. During this time, everything could have become complicated. Either the attachment fell off or the aligner was not fitting properly [30]. However, with the use of DM, these problems are outdated.

The main purpose of the app is to make the work of the doctor and the dental assistants/nurses easier [31]. DM looks at thousands of scans/photos of patients' teeth every day. It sees them from different angles and can compare each photo with another.

In orthodontic treatment with invisible aligners, this sophisticated software is used frequently. It detects an incipient problem early. This type of treatment is used to monitor attachments, buttons, and non-tracking aligners. It will also detect poor hygiene, which will show up as inflamed gums, plaque, and dental calculus. A big plus is that it can also highlight aphthae, gingival recession, and, in certain cases, dental caries [32].

The patient data provided to DM is processed by Artificial Intelligence (AI) and a report is returned to the patient after each scan about their current condition. Depending on the report, patients can get the option of changing their aligners for another set. The standard 10 + 2 protocol can be accommodated by an orthodontist depending on the patient's compliance, hygiene, increment per aligner, biomechanical difficulty of the particular phase of the treatment, and many other aspects.

With the 10 + 2 protocol, the patient can change aligners every 10 days as long as they wear them for more than 22 h a day [33]. If there is a problem, the software will halt the patient on the current aligner for another 2 days before another opportunity for intraoral video-scanning with another evaluation. In such a situation, the reason for the halt is reported to the nurses and doctors at the patient's office. The most common problem is attachment debonding or non-tracking. In cases of attachment debonding, a new template is ordered by nurses upon the decision of the orthodontist about the importance of that particular attachment in the context of the current phase of the treatment as well as the aspect of the estimated date of the new attachment template's delivery in the context of the current wear of the aligner. In some cases, the template is ordered for the next aligner stage, where an appointment is booked in advance without stopping the flow of the treatment. In other cases, a patient is halted on the current aligners until a template arrives and the attachment is rebonded. In those cases, patient's with poor tracking typically improve their compliance, and non-tracking is resolved naturally. In cases of significant non-tracking, it makes no sense to bypass the decision of the DM to manually halt, as such an override would lead to even more tracking. In these cases, an extra aligner of that particular stage can be ordered or even at the −1 stage, and the patient is given a chance to "catch up" with tracking; otherwise, a revision is performed. This means a new scan and photos are made in the office and submitted for a new set of aligners to be manufactured according to the original plan.

Some functions can change these system settings. For example, if the patient has had a splint not fitting on one tooth for a long time and the doctor evaluates this problem as satisfactory, there is a Force GO button. This means that the Artificial Intelligence (AI) will move the patient to the next aligner even if it evaluates the problem.

However, from our point of view, the Force STOP function is missing. It would slow down the treatment of patients who are not 100% compliant with the conditions and rules of treatment.

Figure 3 shows an example of the visual communication of dental assistants/dental nurses, respectively, with the patient, with a message visualizing the problem exported directly from the DM scan. This represents the practical first layer of problem identification, where, depending on the type and severity of the problem, pre-processed with AI telemonitoring recognition, a dental nurse is alerted with treatment issues and fixes them according to the office protocols, alerting doctors only when necessary otherwise handling

communication with or education of the patient [34]. The post-identification problem has three main levels of management:

1.  Semi-automatic AI communication with the patient (e.g., non-ideal dental hygiene with automatic recommendations for improvement);
2.  Digital or face-to-face problem management with the dental assistant (e.g., inadequate dental hygiene or the need to improve the wearing of aligners);
3.  Need for direct decision-making by the dentist, revision of treatment, or direct doctor-patient communication, depending on the severity and nature of the problem detected by the DM app. Or when a clinical alert, setup by the doctor, has been achieved.

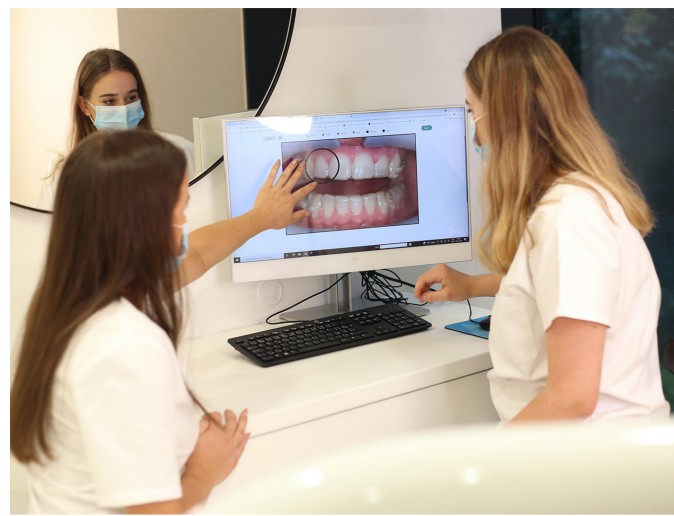

**Figure 3.** Dental assistants/nurses visually communicate problems with the patient represented in the black circle using DM scan exports. AI pre-processing identifies the severity of the problem, and nurses fix the issue(s) by managing patient communication/education; otherwise, doctors are alerted as needed.

Patients receive scan reports, messages from nurses, and scan reminders in the chat section within the DM app. In the past, patients had to keep track of when they changed aligners, how many days they had been wearing the aligner, etc. With this system, they do not have to worry at all about the date they change their aligners. The DM app will remind the patient on the day of the scan to let them know it is scan time. Within a few hours, the patient will be told if they can change their aligner or if they need to stay on that aligner.

Every patient who starts DM is fully trained in the use of the app. It is explained to patients why they need to do ongoing scans. The procedure to use DM is to download the app, create a profile, and perform basic setup on the phone. If everything is done, it goes straight to scanning. The phone is set up in a special dental scan box, which ensures the stability of our photos [35]. Figure 4 shows crucial patient education by nurses directly affecting the effectiveness of telemedicine-enhanced workflow.

Individual scans consist of four photos:

1.  A scan of bitten teeth without aligners;
2.  A scan of the teeth apart, without aligners;
3.  A scan where the patient has his mouth open and the inside of the teeth is scanned, without aligners;
4.  A scan of teeth apart with aligners.

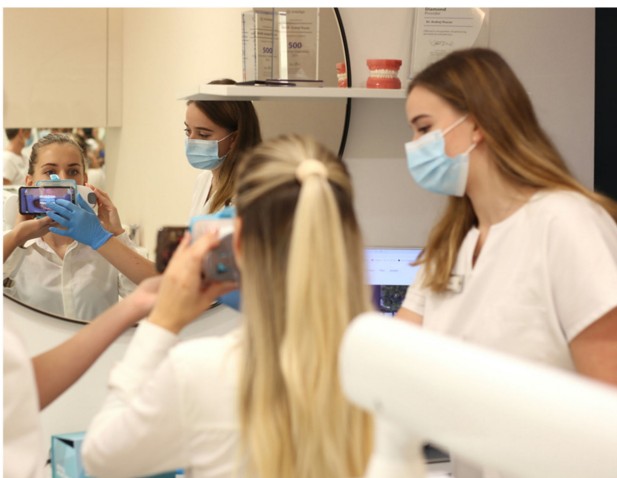

**Figure 4.** The nurse teaches the patient how to scan properly with the Dental Monitoring® (DM) (Dental Monitoring Co., Paris, France) scan box. Demonstration picture.

Each video sequence is processed by Artificial Intelligence (AI) [36] to produce photographs of the teeth taken from multiple angles. The video sequences are examined, and DM generates a conclusion from each one to inform not only the patient but also the clinic staff.

DM facilitates the work of the medical staff. It makes us aware of the problems that trouble the patients. It is unimaginable that we check so many scans and photos of teeth every day as the AI in the DM system manages to do. Because DM detects even the smallest discrepancies, we can recommend treatments to patients to eliminate the problem or reduce the discrepancy.

It is also possible to connect the DM system with the tracking of the treatment plan. One of the newer functions is the setting of treatment target tracking. We can set the AI to alert us when the patient has a corrected intercuspidation, a straight incisor center, or a reduced overbite. We can either set these targets with a date when we expect the result, or we can set the target for a specific aligner. We determine this using the Clincheck plan, where we can see the exact position of the teeth as they should be [37]. Of course, we must always accept slight variations.

In addition to monitoring the treatment with the invisible aligners, we can also use DM to monitor pediatric patients before treatment. It allows us to monitor the development of the patient's teeth without them having to visit us frequently. This way, we can better determine when the right time is to start treatment.

DM has another advantage, which is monitoring patients during retention. After scanning, tooth movement can be evaluated in detail. The report of tooth movement during retention is given in millimeters. This allows us to advise the patient to wear retainers more frequently to prevent further tooth displacement. These two scans are performed only once a month [38–40].

### 2.3. Concept

Orthodontics has seen the development of new diagnostic and therapeutic approaches in recent years, driven by patient expectations and the desire of professionals to improve predictability in treatment planning. Predictability is essential for effective and high-quality treatment [41].

This paper presents a novel concept of engagement of dental assistants for regular AI-telehealth monitoring and evaluation of orthodontic treatment progress with the assistance of AI evaluation of clinical targets set by the orthodontist. Although there is increasing interest in AI applications in digital orthodontics, clear aligners, and sleep dentistry, this paper focuses on specific AI applications and concepts for orthodontics that go beyond

current understanding and clinical solutions and offer a new AI powered digital orthodontic workflow rather than an AI tool for legacy treatment workflows.

Implementing AI in orthodontic treatment monitoring offers the opportunity for quality and effectiveness evaluation and comparison [10,41,42]. The proposed concept introduces DM as a compulsory AI transformation of orthodontic treatment, where the entire treatment journey, including the retention phase, is monitored and compared to millions of other similar treatments worldwide. Providers can be alerted immediately when their patient's progress deviates from the common range of similar tooth movements or when their patient's debonding frequency is above the average for their country or possibly the world. This information is essential for improvement and identifying high-quality providers who can share their workflows and become educators.

AI implementation in Dental Monitoring® (Dental Monitoring Co., Paris, France) gathers big data worldwide, which can become a cornerstone for further improvement of the orthodontic specialty. For dental offices, it will represent a crucial tool for quality and effectiveness assessment.

## 3. Results

Despite the fact that it is impossible to have two exact clinical cases with matching patient compliance and other relevant parameters in orthodontics, it is possible to estimate the impact of technology in proportion. Aligner treatment of comprehensive treatment on patients before and after implementation of the DM workflow resulted in a change in the number of refinements in the patient, in-person visits to a dental clinic, as well as the number of forced revisions per patient. Average comprehensive treatment refinements per patient represent the average number of requested additional aligners (AA) after the patient finishes the prescribed series of manufactured aligners, typically to finish the minor details. Multi-phase treatments, where another set is requested as another planned phase of treatment, were not included. For example, after the phase of correcting the crossbite, another phase of aligners is designed and manufactured to manage remaining problems after occlusal shift (this was not counted as AA).

The number of in-person appointments per year of treatment (52 weeks) for patients before DM was set up by the clinician, was found to be approximately 5–7 weeks, depending on the complexity of a given phase of treatment and the degree of patient compliance. Only in patients with DM did this depend on the DM visual and AI feedback.

Forced treatment revision per patient represents a necessity and is evaluated subjectively by a clinician to determine that the discrepancy in aligner fit has already exceeded a reasonable threshold and a new series of aligners needs to be manufactured.

Average treatment length in weeks per comprehensive case, with only treatments longer than 52 weeks included. Although we report here the average duration of treatment, this figure also includes surgical treatments in which the time to coordinate treatment with orthognathic surgery is included, and therefore this time figure contains undue bias. As well as the comprehensive treatments that were under the defined threshold of 52 weeks with DM, these were excluded, and this value shall be considered with its significant limitations and bias.

The impact of DM implementation on the proportion of nurse/dental assistant main activities (chair assistance and communication) per day is shown in Figure 5.

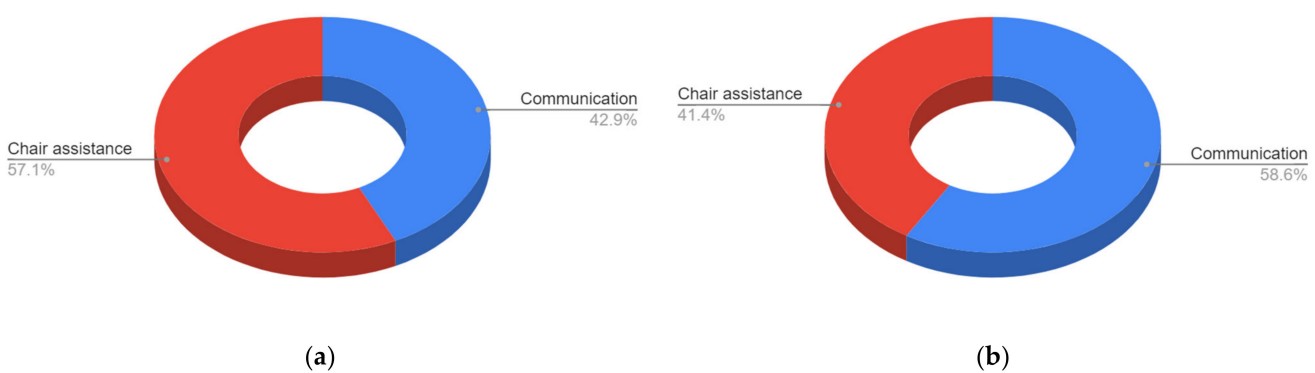

(**a**)                   (**b**)

**Figure 5.** The proportion of direct chair assistance from the nurse/dental assistant was reduced in comparison to the amount of communication after DM implementation. Additional communication with the patient in the dental office was brought in by DM, as it often prevented the patient from moving to the next aligner when a problem was observed. (**a**) Prior to DM implementation, dental nurses mainly communicated with patients via email and SMS, mostly regarding appointment times and dates; (**b**) DM has shifted the amount of necessary communication that can be semiautomated later by AI.

## 4. Discussion

This paper describes the digital concept of an AI-powered telehealth workflow with novel characteristics for dental assistant and nurse responsibilities. The advantages described correlate with the current scientific reports on Dental Monitoring® (Dental Monitoring Co., Paris, France). It is promising tele-dentistry software that accurately tracks a patient's clear aligner treatment. It brings many benefits to orthodontic practice, but also many issues and new responsibilities. As shown, it does not necessarily represent less chair time in the short term, although it increases the treatment effectiveness by a significant increase in permanent control over the treatment [43].

The introduction of DM can significantly change the workflow in dental practices. It can lead to a significant reduction in the average number of treatment refinements and, thus, a reduction in the overall duration of treatment. However, it does not necessarily reduce the average number of appointments per year, as the AI in DM quickly detects dropped attachments from patients' home video scans or damage to other orthodontic accessories, requiring nurses to call the patient back to the dental office to reattach in this situation. Implementing DM into workflows may even double the number of forced revisions to treatments for non-compliant patients. In orthodontic workflows without dental supervision, such cases were often overlooked and arrived at the final scheduled aligners with major fitting discrepancies associated with comfort and speech impairments and a further reduction in patient compliance.

By using traditional methods, dentists face unnecessary issues such as a prolonged time between each monitoring session or wearing an aligner, which has no added value to our treatment, but because there is no short-term monitoring, this cannot be ascertained.

For many people, the discovery and spread of the printing press was an overhyped discovery. However, the moment of diffusion of information and skills that had previously been trapped and slowed down by the need to rewrite a particular book was a turning point for humanity. In a similar way, the first forms of autonomously trained broad-spectrum AI at the frontier of AGI today suggest a similar moment, where a hyper-intelligent tool trained in one place will be able to be deployed in many different places in many different services, with unprecedented impact on many professions, including nurses in dental clinics.

The limitation of this preliminary report is the pace at which the field is changing. While analytical AI has not yet penetrated all tools in dentistry, generative AI is already introducing forms of communication with patients.

The current practical limitations and drawbacks of DM, as we discovered, are imperfections in its application during daily use. Although AI mainly focuses on attachment tracking and aligner tracking if attachments change during treatment (some are removed, others are added), it happens that it cannot track them, does not report dropped attachments, or reports them as dropped attachments [38]. This means that we cannot rely 100% on AI alone. Therefore, the scans also need to be checked by health professionals/doctors.

In addition, after comparing individual scans, we concluded that AI judges some photos more harshly than others. If a patient has a badly fitting aligner in a certain place, the DM app will stop it and give them two days to fix the problem. In some scans, we noticed significant non-tracking of the aligners, but the artificial intelligence evaluated that the condition was fine [39]. Conversely, for some patients, when there is minimal non-tracking, the AI will stop them for longer periods.

An advantage of DM for frequently traveling patients lies in their phones. The comfortable usage of their mobile phone app provides an adequate image of patients progress without using the DM scan box. The retractor is also used in the standard scanning procedure, helping to keep the lips and cheeks out of the frame. Without a scan box, we are comfortable scanning by holding the phone in one hand and turning our heads to the side [40].

Another advantage of the DM app is direct communication between office and patient to provide orthodontic care and patient education without physical contact with the dentist.

Although DM does not necessarily decrease or even increase the frequency of appointments, the Gaussian distribution of these appointments is concentrated in the first weeks of treatment, when attachment debonding is more common, which is related to patients learning proper habits of using aligners, including correct removal routines and others. Although a more controlled treatment requires more attention initially, reducing the overall treatment time with better finishing and stability should be the dentist's priority.

In terms of shifting the proportions between communication and the hands-on work of the dental assistant, DM opens a new communication channel in the dental clinic that should not be ignored. Patients observed at the DM are assessed not only by the AI but also by orthodontists. The AI makes it possible to strictly stop the treatment progress in cases of insufficient fit of aligners, insufficient hygiene, detachment of attachments, or other problems defined by the orthodontist. Patients have a natural tendency to communicate these issues with the office. This is leading to a significant increase in communication about identified clinical problems and treatment recommendations by phone, chat, or directly through the messaging system of DM. Insightful patients who understand the problem are always better than those who must blindly obey instructions. Most of this new burden of additional communication is handled by dental assistants and nurses, who consult with physicians as needed. Frequently asked questions from patients can already be answered semi-automatically by other AI algorithms, and it is hard to imagine these digital AI-enabled workflows without them. Already, the way staff communicate with patients can be automatically used to train a dedicated AI communication module that would be available to the clinic through available AI algorithms such as www.personal.ai and the like.

The DM software critically evaluates even the slightest deviation of the aligners and allows the patient to sit on the current number of aligners until the aligners fit perfectly. As the aligners fit better, a proper force system is applied to the teeth. The software requires patients to scan their teeth with each aligner and maintains control of the entire treatment. Doctors are alerted to monitor problematic patients for a lack of tracking, missing fixtures, inadequate dental hygiene, and other issues that can negatively impact treatment outcomes, making it easier to take care of patients who need acute intervention.

On the other hand, perfect cooperation from the patients is required for the software to work accurately. Fortunately, we find only in a few cases that patients forget or ignore the set time of the scans or make the decision from DM to wear the same aligner longer if needed. In these cases, the treatment may fail more easily [10].

The proper execution of dental scans is crucial for accurate evaluation. Incorrect holding or phone angulation, as well as insufficient soft tissue retraction, can lead to errors in assessment. Additionally, precise scanning procedures are necessary for the proper detection of lingual side attachments and auxiliaries, which can be challenging to control. Despite these challenges, DM has been shown to be a valuable tool in improving orthodontic treatment outcomes. However, there are limitations, such as difficulty in accurately assessing hygiene due to the reliance on light intensity and the potential misperception of healthy gingiva as inflamed.

The integration of AI in dental clinics has the potential to revolutionize the way dental assistants work by automating tasks such as scheduling appointments, taking patient histories, and providing patient education. Furthermore, AI can be utilized to analyze patient data and create more accurate diagnoses and treatment plans, as well as improve the accuracy of dental procedures such as the detection of cavities and the creation of more detailed images of teeth and gums. Additionally, AI can be used to enhance the patient experience by using chatbots to answer questions and provide personalized advice.

The clinical penetration of these technologies has been accelerated by the global COVID-19 pandemic and technological breakthroughs in AI and personal smartphones. The future of AI in dentistry is promising and holds significant potential for improving both the efficiency and effectiveness of dental treatment.

The integration of AI in orthodontic practice is changing the role of dental nurses in several ways:

1. Automation of tasks: AI can automate tasks that dental nurses currently perform, such as scheduling appointments, taking patient histories, and providing patient education. This can free up time for dental nurses to focus on more complex tasks or to interact with patients in a more meaningful way;

2. Improved accuracy: AI can be used not only to analyze patient data or create more accurate diagnoses and treatment plans. It can also predict complications, keep an eye on allergies, or provide a nonstop monitor and alert system in harmony with telehealth solutions. This can lead to improved patient outcomes and increased efficiency in the clinic;

3. Personalized treatment plans and communication: AI can be used to create personalized treatment plans based on a patient's individual needs and perform more personalized communication depending on patient individuality and shown communication preferences. This can lead to better patient outcomes and increased patient satisfaction;

4. Improved patient experience: AI-powered chatbots that will be learning from nurses' answers to patient questions will provide personalized advice;

5. The combination of telemedicine and AI provides software solutions for orthodontic treatment that allow some parts of orthodontic treatment to be performed in locations other than the dental office. This can lead to further disruption of established clinical care workflows.

In general, the use of AI-powered telemedicine solutions in dentistry has transformed the work of dental assistants by making them more efficient and accurate and allowing them to focus on more complex tasks and patient interactions. One such more complex task that requires a combination of clinical interaction, reasoning, and physical intervention can be considered the personalization of treatment in terms of personalized orthodontic appliances. In the past, it was unthinkable that some procedures could be performed without the complex theoretical knowledge and manual skills of laboratory technicians. In the near future, many boundaries will fall, and AI algorithms will allow nurses and dental assistants to participate in procedures previously reserved for highly specialized technicians, such as



CBCT AI segmentations or AI processing of intraoral scans to prepare designs for individualized appliances in preparation for the orthodontist. For example, the design of 3D-printed personalized composite appliances can be semi-automated, and the role of nurses and dental assistants can become much more universal with AI empowerment [4,44–46].

Automated processing of medical data in dental practices by AI can detect much more than is currently expected. From automated cephalometric radiography to growth and ageing predictions, age determination, or even future forensic applications [47,48].

In fact, nowadays dental nurses' workflow is changing profoundly. Therefore, future research is needed to combine AI technology with the use of recently introduced low noise instruments [49], experimental computerized devices [50], and new preventive strategies [51] to increase workplace security and allow a better experience for patients. The authors of this article believe that personal care by nurses and dental assistants will become more valuable as various AI chatbots become widely available and human attention becomes more precious and expensive. The focus of human nurses and dental assistants will be on more complex tasks that will not be stereotypical and will require high efficiency in terms of empathy and complex practical thinking, followed by actions that engage the physical body.

This paper suggests that the implementation of AI, particularly AI-powered language models and teledentistry, is changing the role of dental assistants and nurses in orthodontic practices. The study presents evidence of the benefits and drawbacks of automated assessments of orthodontic treatment progress and the impact of AI on personalized care. The research concludes that AI has the potential to improve dental practice by providing more precise and personalized treatment and creating new roles and responsibilities for trained medical professionals. However, the implementation of AI in dentistry raises new ethical and legal issues for dental practices [10,11,45,52–54].

In the context of other evidence, this paper adds to the growing body of research that suggests that AI has the potential to transform orthodontic healthcare, not only in the fields of intelligence-based remote monitoring of clear aligner therapy [55], crowding categorization from radiographs [56], AI-automated cephalometrics [57], or CBCT AI segmentation [54].

The use of AI in dentistry is still in its early stages, but this study provides evidence that AI can improve orthodontic treatment workflows and create new roles for dental assistants and nurses.

The implications for future research are significant, as this workflow can also be applied to non-aligner therapies. In non-aligner therapies, the DM does not evaluate the tracking of aligners but all other previously mentioned features as well as the debonding of the brackets, their damage, or releasing the orthodontic wire. Further research is needed to explore the potential benefits and drawbacks of AI in dentistry, particularly in other areas of dental practice beyond orthodontics. In addition, research is needed to address the ethical and legal issues raised by the implementation of AI in dentistry. These include issues related to privacy, patient consent, and the regulation of AI in healthcare. In summary, this paper highlights the potential of AI to transform healthcare and emphasizes the need for further research to fully understand its implications.

## 5. Conclusions

The combination of AI and teledentistry has become a clinical reality, changing the workflow of modern orthodontic practices and the role of dental assistants. This requires a rethinking of the old paradigm of orthodontic care, providing more effective and affordable treatment with higher levels of personalization and clinical effectiveness. The new role of dental nurses empowered by AI focuses on communicating with patients and filtering reports from the AI telemonitoring system for clinicians.

Implementing AI-powered telemedicine tools like DM into the orthodontic workflow can reduce the overall duration of treatment and provide better visual control over treatment, with the ability to set up AI alerts for individual clinical or other conditions. Regular

visual control of the intraoral situation and aligner fit could lead to more frequent patient visits to the dental clinic for debonded attachment or other issues.

AI in combination with telehealth solutions can automate tasks for dental nurses, freeing up time for more complex tasks or patient interaction. It can also improve patient outcomes and efficiency by analyzing data, predicting complications, creating personalized treatment plans and communication, and offering personalized advice through AI-powered chatbots. Additionally, it can disrupt established clinical care workflows by allowing orthodontic treatment to be performed remotely.

The near future will bring further significant disruptions, with new generations of powerful text-generating AI, such as ChatGPT, becoming more available and economical than human care or communication. This will become commonplace and the domain of low-cost medical care. It will be the natural right of patients to know if the communication on the side of the dental office is human or artificial intelligence (AI).

**Author Contributions:** Conceptualization, A.T., S.H., M.S., R.U. and J.S.; methodology, A.T.; software, A.T.; validation, M.L., A.T., S.H., M.S., R.U. and J.S.; formal analysis, A.T.; investigation, S.H.; resources, A.T.; data curation, A.T. and J.S.; writing—original draft preparation, A.T., S.H., M.S., R.U., A.T., M.L. and J.S.; writing—review and editing, A.T., S.H., M.S., R.U. and J.S.; visualization, S.H.; supervision, A.T.; project administration, A.T.; funding acquisition, A.T. All authors have read and agreed to the published version of the manuscript.

**Funding:** This work was supported by the Slovak Grant Agency for Science (KEGA Thurzo)—grant No. 054UK-4/2023 and APVV-22-0381.

**Institutional Review Board Statement:** Not applicable.

**Data Availability Statement:** Not applicable.

**Acknowledgments:** The authors gratefully acknowledge the technological support of the digital dental lab infrastructure of 3Dent Medical Ltd. company (Bernolákovo, Slovakia) as well as dental clinic Sangre Azul Ltd. (Bratislava, Slovakia), and we would like to express our gratitude to ChatGPT 4's ability to understand and interpret complex concepts, which greatly aided in the final revision and refinement of our research and analysis.

**Conflicts of Interest:** The authors declare no conflict of interest.

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
