# Peer review of "The New Role of the Dental Assistant and Nurse in the Age of Advanced Artificial Intelligence in Telehealth Orthodontic Care with Dental Monitoring: Preliminary Report"

_applsci, doi:10.3390/app13085212_

Round 1

Reviewer 1 Report

The article is interesting, but it's necessary to implement it. 

Tthe introduction section needs to be implemented, read and add the following articles in the introduction:

1. Impellizzeri A, Horodinsky M, Barbato E, Polimeni A, Salah P, Galluccio G. Dental Monitoring Application: it is a valid innovation in the Orthodontics Practice? Clin Ter. 2020 May-Jun;171(3):e260-e267. doi: 10.7417/CT.2020.2224. PMID: 32323716.

2. Impellizzeri A, Horodynski M, De Stefano A, Palaia G, Polimeni A, Romeo U, Guercio-Monaco E, Galluccio G. CBCT and Intra-Oral Scanner: The Advantages of 3D Technologies in Orthodontic Treatment. Int J Environ Res Public Health. 2020 Dec 16;17(24):9428. doi: 10.3390/ijerph17249428. PMID: 33339197; PMCID: PMC7765620.

The materials and methods section lacks some details. How were the patients selected? What are the inclusion and exclusion criteria? What statistical analysis was done to establish the significance of the sample?

In the results section add the part relating to statistical analysis.

Author Response

Reviewer 1

Comments and Suggestions for Authors

The article is interesting, but it's necessary to implement it. 

Tthe introduction section needs to be implemented, read and add the following articles in the introduction:

  1. Impellizzeri A, Horodinsky M, Barbato E, Polimeni A, Salah P, Galluccio G. Dental Monitoring Application: it is a valid innovation in the Orthodontics Practice? Clin Ter. 2020 May-Jun;171(3):e260-e267. doi: 10.7417/CT.2020.2224. PMID: 32323716.
  2. Impellizzeri A, Horodynski M, De Stefano A, Palaia G, Polimeni A, Romeo U, Guercio-Monaco E, Galluccio G. CBCT and Intra-Oral Scanner: The Advantages of 3D Technologies in Orthodontic Treatment. Int J Environ Res Public Health. 2020 Dec 16;17(24):9428. doi: 10.3390/ijerph17249428. PMID: 33339197; PMCID: PMC7765620.

The materials and methods section lacks some details. How were the patients selected? What are the inclusion and exclusion criteria? What statistical analysis was done to establish the significance of the sample?

In the results section add the part relating to statistical analysis.

Submission Date 24 February 2023
Date of this review 08 Mar 2023 11:47:19

Dear Reviewer,

Thank you for your recommendations and comments. We have presented the two papers you recommended by Impellizzeri et al. in the introduction. Lines 137-144

Because our work is a review, we do not address the method of patient selection or the statistical analysis, as none was used.

All changes made are tracked and visualised with the "Tracking Changes MS Office Tool".

Kind regards

authors

Reviewer 2 Report

Dear Authors,

congratulations on well written manuscript. There are some minor language/style mistakes (e.g. line 50, 117, 494).
Additionally, I believe it is worth mentioning in the title that you investigated a specific tool (without the corporate name) apart from the nurses role it will draw more attention.

Best regards
Zuzanna Nowak

Author Response

Dear Authors,

congratulations on well written manuscript. There are some minor language/style mistakes (e.g. line 50, 117, 494).
Additionally, I believe it is worth mentioning in the title that you investigated a specific tool (without the corporate name) apart from the nurses role it will draw more attention.

Best regards
Zuzanna Nowak

Submission Date

24 February 2023

Date of this review

06 Mar 2023 10:10:14

Dear Reviewer,

Thank you for your compliment and remarks. We have corrected the syntax errors and proofread the paper again. We have also improved the Title according to your suggestions; we agree with you. We have changed it to: “The New Role of the Dental Assistant and Nurse in the Age of Advanced Artificial Intelligence in Telehealth Orthodontic Care with Dental Monitoring

All changes made are tracked and visualised with the "Tracking Changes MS Office Tool".

Kind regards

Authors

Reviewer 3 Report

This paper explores the impact of Artificial Intelligence (AI) on the role of dental assistants and nurses in orthodontic practises, as there is a gap in understanding the currently evolving impact on orthodontic treatment workflows. The introduction of AI-language models such as ChatGPT is changing patient-office communication and transforming the role of orthodontic nurses. Teledentistry is now in orthodontics profoundly dependent on AI implementation. This paper presents the proof of a novel concept of AI-powered orthodontic workflow providing new responsibilities for an

orthodontic nurse. It also provides a report of an assessment of such workflow in an orthodontic practice that uses an AI solution called Dental Monitoring from a period of three years. The paper evaluates the benefits and drawbacks of daily automated assessments of orthodontic treatment progress and the impact of AI on personalized care and the new role of a dental assistant. The paper concludes that AI will improve dental practice through more precise and personalized treatment,

bringing new roles and responsibilities for trained medical professionals, but raising new ethical and legal issues for dental practices.

The work deals with important aspects of modern dentistry and, in my opinion, has a very large impact on the development of artificial intelligence techniques in the dental industry.

Author Response

This paper explores the impact of Artificial Intelligence (AI) on the role of dental assistants and nurses in orthodontic practises, as there is a gap in understanding the currently evolving impact on orthodontic treatment workflows. The introduction of AI-language models such as ChatGPT is changing patient-office communication and transforming the role of orthodontic nurses. Teledentistry is now in orthodontics profoundly dependent on AI implementation. This paper presents the proof of a novel concept of AI-powered orthodontic workflow providing new responsibilities for an

orthodontic nurse. It also provides a report of an assessment of such workflow in an orthodontic practice that uses an AI solution called Dental Monitoring from a period of three years. The paper evaluates the benefits and drawbacks of daily automated assessments of orthodontic treatment progress and the impact of AI on personalized care and the new role of a dental assistant. The paper concludes that AI will improve dental practice through more precise and personalized treatment,

bringing new roles and responsibilities for trained medical professionals, but raising new ethical and legal issues for dental practices.

The work deals with important aspects of modern dentistry and, in my opinion, has a very large impact on the development of artificial intelligence techniques in the dental industry.

Submission Date

24 February 2023

Date of this review

09 Mar 2023 14:12:23

Dear Reviewer,

Thank you for your compliment and remarks.

Kind regards

Authors

Reviewer 4 Report

Previous studies have investigated artificial intelligence in orthodontic smart application for treatment coaching through 3 years follow-up. Through literature search, the reviewers have found that there were a few articles with similar research, but almost no long-term clinical observation was found. For the purpose of research, the current study has certain innovation and clinical value, but some serious problems have been found in the manuscript. Some are detailed as follows for the author's consideration.

In the latter part of the Introduction, it is suggested to further analyze the limitations and deficiencies of the current relevant research, so as to highlight the necessity of this study.

Additionally, it is suggested to mark the structure of each part with arrows on Figure 2 in the Material part.

Next to Figure 4, the author is suggested to further supplement the example schematic diagram or pattern diagram of the 4 photos, so that readers can understand more intuitively.

Table 1 should be replaced with a three-wire table.

Table 2 mentioned in the Discussion does not appear in the manuscript. Besides, it is suggested that the author increase the analysis of the results of this study and elaborate the advantages of AI application in orthodontic treatment.

Author Response

Previous studies have investigated artificial intelligence in orthodontic smart application for treatment coaching through 3 years follow-up. Through literature search, the reviewers have found that there were a few articles with similar research, but almost no long-term clinical observation was found. For the purpose of research, the current study has certain innovation and clinical value, but some serious problems have been found in the manuscript. Some are detailed as follows for the author's consideration.

In the latter part of the Introduction, it is suggested to further analyze the limitations and deficiencies of the current relevant research, so as to highlight the necessity of this study.

Additionally, it is suggested to mark the structure of each part with arrows on Figure 2 in the Material part.

Next to Figure 4, the author is suggested to further supplement the example schematic diagram or pattern diagram of the “4 photos”, so that readers can understand more intuitively.

Table 1 should be replaced with a three-wire table.

Table 2 mentioned in the Discussion does not appear in the manuscript. Besides, it is suggested that the author increase the analysis of the results of this study and elaborate the advantages of AI application in orthodontic treatment.

Submission Date

24 February 2023

Date of this review

06 Mar 2023 13:04:12

Dear Reviewer,

Thank you for your comments.

We have revised the chapter Introduction.

We have also redrawn Figure 2 with the arrows you suggested.

Regarding your recommendations to revise Figure 4 and Table 1, we have decided to leave them as they are, as they conform to the prescribed template of the Applied Sciences journal.

We have corrected the reference to Table 2 in the discussion, which did not appear in the text.

We have improved the interpretation of the information in the results chapter of this review, although obviously less on data analysis. It is a review paper and in the introduction we had elaborated more on the benefits of AI application in orthodontic treatment, as you recommended.

All changes made are tracked and visualised with the "Tracking Changes MS Office Tool".

Kind regards

Authors

Round 2

Reviewer 1 Report

Well done! The article is completed and well described. 

Author Response

Thank you!

Reviewer 4 Report

Agree with the revisions made by the author and support for publication in Applied Sciences.

Author Response

Thank you!